# Salvage Hip Arthroplasty in Nail Failure: A Systematic Review

**Salvatore Pantè** [1,*]**, Lorenzo Braconi** [1]**, Umberto Cottino** [2]**, Federico Dettoni** [2] **and Roberto Rossi** [1,2]

[1]  Department of Surgical Sciences, University of Study of Turin, Via Po 8, 10100 Turin, Italy; lorenzo.braconi95l@gmail.com (L.B.); rossir@fastwebnet.it (R.R.)

[2]  Department of Orthopedics and Traumatology, AO Ordine Mauriziano, Largo Turati 62, 10128 Turin, Italy; umberto.cottino@gmail.com (U.C.); federicodettoni@tin.it (F.D.)

*  Correspondence: salvatorepante.md@gmail.com

**Abstract:** Background: Femoral nailing is a largely widespread procedure in the elderly population, and the number of these surgeries is rising. Hip arthroplasty is a salvage procedure performed to improve function of the hip after femoral nail failure. The aim of the study is to evaluate functional outcomes, complications and survivorship in hip arthroplasty after femoral nail failure. Methods: A systematic review of three databases (Pubmed, Embase and Cochrane) was performed using the PRISMA 2020 guidelines. After selection, four studies met the inclusion criteria, and 483 treated hips have been analyzed (476 total hip arthroplasties, 7 partial hip arthroplasties). Results: The median value of Harris Hip Score (HHS) after salvage treatment was 86.1. The main indications for salvage treatment were osteoarthrosis, avascular necrosis of the femoral head and instability of the hip. Complications are more frequent than in primary total hip arthroplasty, in particular aseptic loosening and dislocation. Good outcomes have also been achieved using revision-type stems and proximal femoral replacements (PFR). Conclusions: Conversion total hip arthroplasty is confirmed as the optimal treatment for femoral nail failure in the elderly population. Cemented or hybrid total hip arthroplasties have better outcomes than uncemented total hip arthroplasties, and the use of different types of implants widens the possible approaches to surgery in restoring the biomechanics of the hip and increases the satisfaction of patients.

**Keywords:** femoral nail failure; conversion arthroplasty; salvage THA; hip fracture



## 1. Introduction

Lateral hip fractures in the elderly population are among the most common orthopedic injuries, with more than 250,000 cases every year in the United States [1]. Burge et al. [2] estimated the incidence of this type of fracture to increase by 50% by 2025. According to literature, 90% of lateral hip fractures occur in geriatric patients [3], usually as a result of a low-energy trauma, such as a fall from a standing height [4]. Extramedullary and intramedullary fixation and hip arthroplasty are the available surgical alternatives.

Nowadays, intramedullary nailing is becoming the first-choice treatment worldwide [5,6]: based on the American Board of Orthopedic Surgery examination database, there is an ongoing trend towards treating lateral hip fractures with cephalo-medullary nails (CMN), increasing from 3% in 1999 to 67% in 2006 [7]. Other sources certificate that femoral nailing is rising over the years (from 28% in 2006 to 64% in 2011 in South Korea and from 65% to 80% in 2017 in Denmark) [8,9].

Femoral nailing has several advantages. The closer positioning of the implant to the femoral mechanical axis results in less implant strain, a shorter lever arm, increasing stability [10], less periosteal stripping and soft tissue damage, a shorter hospital stay and operation time, fewer blood transfusions, improved postoperative walking and decreased incidence of leg length discrepancy [11,12]. Even though proximal femur fracture fixation with cephalo-medullary nails generally yields positive outcomes, fixation failures could affect a patient's recovery and functional prognosis at any point after surgery. Breakage of

the intramedullary nail is an uncommon complication, and it is related to fracture healing failure that can be both depending on the patient (osteoporosis and associated morbidity) or surgery (incorrect implant positioning or poor fracture reduction) [13]. Most of these failures occur at the level of the fixation of the cervical screw through the nail, resulting in cut-out or cut-through. [14–16]. Of paramount importance is to rule out infection every time there is an unhealed fracture or a fixation failure, so much that infection can be a life-destroying event for the patients and can radically change the approach to the revision surgery [17–19]. Conversion arthroplasty is generally accepted as a salvage option in femoral nail failure in elderly people, especially considering the poor bone stock after the implant removal and the femoral head condition [20–22].

The purpose of this systematic review is to evaluate functional outcomes, complications and survivorship of hip arthroplasty after cephalo-medullary nail failure in proximal femur fractures of the elderly.

## 2. Materials and Methods

### 2.1. Search Strategy

A systematic literature search was performed in accordance with the Preferred Reporting Items for Systematic Reviews and Meta-Analyses (PRISMA) guidelines published in 2020 [23] (Figure 1). Two reviewers independently searched 3 online databases (PubMed, Cochrane and EMBASE) using the following keywords: (Salvage total hip arthroplasty) OR (Salvage endoprosthesis) OR (conversion total hip arthroplasty) OR (conversion endoprosthesis) OR (hip nail failure) OR (pertrochanteric nail failure) OR (subtrochanteric nail failure) OR (Femur Failed Osteosynthesis) NOT (Primary total hip arthroplasty) NOT (primary endoprosthesis). All articles published until 17th March 2023 were included. The protocol for this systematic review is registered and available on the Prospero database: CRD42023407695. Institutional review board approval was not required due to the type of study, performed on public paper data.

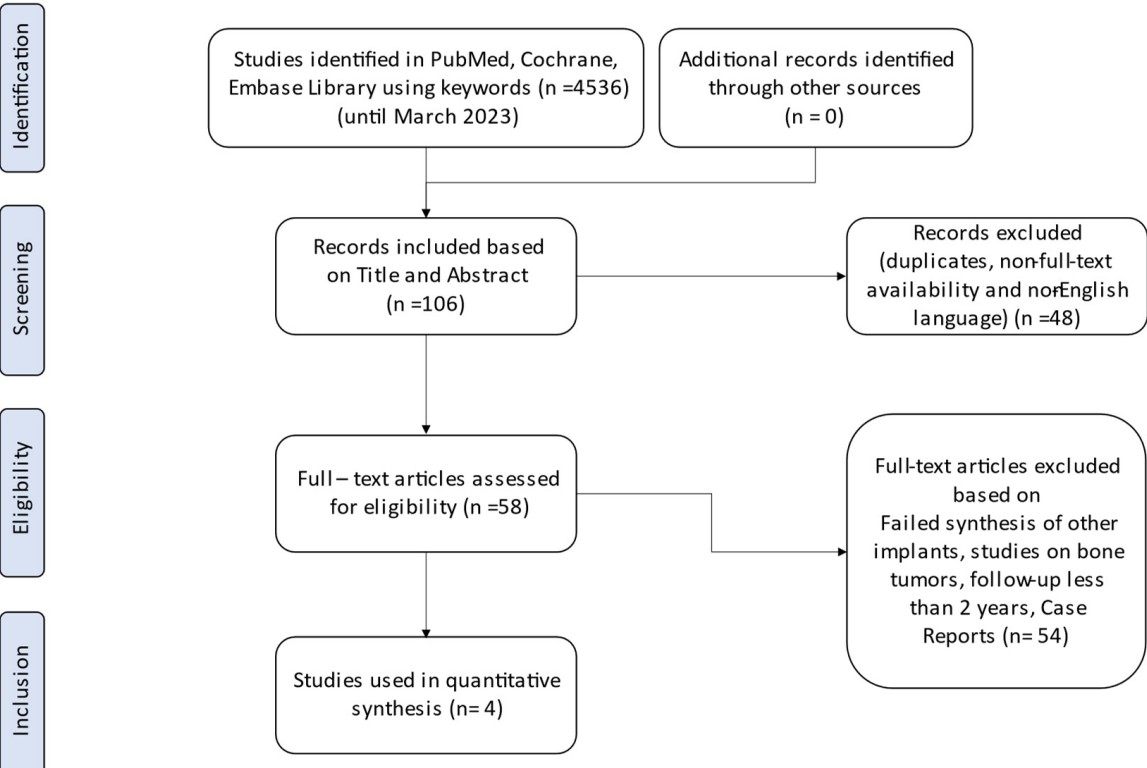

**Figure 1.** PRISMA flowchart.

*2.2. Eligibility Criteria*

Eligible studies were included with the following criteria: (1) level I to IV evidence, (2) English language, (3) failed femoral nail to treat proximal femoral fractures, (4) clinical outcomes reported, (5) minimum 2 years follow-up and (6) full-text papers. Exclusion criteria were as follows: (1) pathological femur fractures, (2) primary hip arthroplasty and (3) failure of other fixation systems.

*2.3. Literature Appraisal*

Two authors screened the title, abstract and full text of the selected studies to determine eligibility. The screening was revised by the senior reviewer. The final decision on inclusion was made on the full-text article.

*2.4. Data Extraction*

Eligible studies were used to extract relevant data including author, year of publication, sample size, study design, level of evidence and surgical procedure. The primary outcome measure in our systematic review was Harris Hip Score after the salvage procedure. Secondary outcomes included postoperative complications, causes of revision and survivorship rate.

**3. Results**

*3.1. Search Results*

The search yielded 4536 results, and 106 studies were selected by title. A total of 23 were discarded due to the exclusion criteria, and 25 were duplicates, having at the end 58 studies. No additional records were included. After screening for eligibility criteria, four articles were selected and included in this study (Figure 1) [24–27], and all reported on the outcomes and complications following conversion arthroplasty after femoral nail failure in intertrochanteric femoral fractures.

*3.2. Characteristics of the Included Studies*

All studies were retrospective, and the total number of hips treated was 483, with the average age of the patients being 69.2 years (range from 56 to 93 years). The male–female ratio was 244/239 and the follow-up range was from 2 to 10 years (Table 1). Three studies [24–26] reported the fracture classification using the AO/OTA classification and were all classified as 31A1, 31A2 or 31A3. The only fixation method was femoral nailing, and the type of the removed nail was reported in all the selected studies.

The causes of failed fixation were reported in all cases, and they can be classified as instability (necrosis of the femoral head, pseudoarthrosis, coxarthrosis and periprosthetic fracture), mechanical failure (cut-out, cut-through, nail migration and nail breakage) or instability and mechanical failure: 199 for instability, 158 for mechanical failure and 136 for both. The failure occurred in a wide span of time according to the cause of failure, from lag-screw cut-out to secondary coxarthrosis. The time from the failure to the salvage procedure was variable and ranged from 7 months to more than 2 years.

The total amount of 483 cases included 476 conversion total hip arthroplasties and 7 conversion partial hip arthroplasties. The surgical approach used for the salvage procedure was not reported in two studies [25,26], while one reported a postero-lateral approach [27] and the other reported an extended direct lateral approach on previous scars [24].

**Table 1.** Studies included and patients data.

| Author | Year of Publication | Study Design | N° Patients | Age | M/F | Salvage Procedure | Average Follow-Up |
|---|---|---|---|---|---|---|---|
| Godoy-Monzon et al. [27] | 2019 | Retrospective analysis of the outcomes of conversion THA using a modular stem following a CMN failure | 28 | 72.7 (62–83) | 11/17 | THA = 28 PHA = 0 | 64 months |
| Yu et al. [26] | 2020 | Retrospective analysis of the outcomes of failed PFNAs converted to a UTA or CTA in elderly individuals | 198 | 66 (62–71) | 92/106 | THA = 198 PHA = 0 | 65 months |
| Innocenti et al. [24] | 2021 | Retrospective analysis of the outcome and complication rate in a group of elderly patients who underwent PFR as a salvage treatment after CMN mechanical failures | 21 | 85.3 (78–93) | 16/5 | THA = 14 PHA = 7 | 37 months |
| Shi et al. [25] | 2022 | Retrospective analysis of the outcomes of the conversion of failed PFNAs to uncemented versus hybrid THAs in the elderly population | 236 | 66.4 (65–71) | 125/111 | THA = 236 PHA = 0 | 120 months |

THA: total hip arthroplasty; CMN: cephalo-medullary nail; UTA: uncemented total hip arthroplasty; CTA: cemented total hip arthroplasty; PRF: proximal femoral replacement; PFNA: proximal femoral nail anti-rotation; PHA: partial hip arthroplasty.

The implant and type of fixation in revision surgery were reported at various levels of detail in each paper. The uncemented total hip arthroplasties were 214, the hybrid ones were 134, the cemented ones were 100 and the partial hip arthroplasties were 7; in one paper [27] were reported 28 implants with various fixation types, but it was not possible to understand the combination of single elements. All the specific types of prosthetic implants used for the salvage procedure were reported in all included studies.

Various scoring systems were used in the included studies to assess functional outcomes, and the most common was the Harris Hip Score (Table 2).

The MINORS scoring system [28] was used to determine quality rating for the articles included (Table 3). The mean quality rating for all studies included was 15.5 points (range 11–21) of a possible 24 points for comparative studies, which shows that the studies are, on average, of good methodological quality.

**Table 2.** Fracture type, cause of failure, implant and surgery characteristics.

| Author | Initial Fracture (AO/OTA) | Femoral Nail | Cause of Failure | Time from Failure to Arthroplasty | Implant | Surgical Approach |
|---|---|---|---|---|---|---|
| Godoy-Monzon et al. [27] | Not specified | PFN—Synthes (13)<br>TFN—Synthes (9)<br>Gamma3—Stryker (6) | Osteoarthritis (11)<br>AVN/head collapse (7)<br>Pseudarthrosis (5)<br>CMN migration (4)<br>CMN breakage (1) | 12.6 months (range 7–20) | Uncemented MFSs stem (23) or Cemented MFSs stem (5)—Waldemar Link<br><br>Uncemented modular cup (24) or Cemented all-poly cup (4)—Waldemar Link | Posterolateral approach |
| Yu et al. [26] | 31A1 (35)<br>31A2 (118)<br>31A3 (45) | PFNA—Smith&Nephew (198) | Instability (48)<br>Mechanical failure (45)<br>Both (105) | <1 year (32)<br>1–2 years (109)<br>>2 years (57) | UTA (98)<br>Corail uncemented stem (DePuy) and Reflection uncemented cup (Smith&Nephew<br><br>CTA (100)<br>Exeter stem and cup (Stryker) | Not specified |
| Innocenti et al. [24] | 31A2.2 (3)<br>31A2.3 (9)<br>31A3.3 (9) | PFNA—Synthes (9)<br>CHIMAERA—Orthofix (6)<br>Gamma3—Stryker (4)<br>INTERTAN—Smith&Nephew (2) | Nail breakage (17)<br>Lag screw cut-out (4) | 9.6 months (range 8–14) | Modular DM Delta TT cup (14)—Lima or Bioplar head (7)—VarioCup-Link<br><br>Megasystem C-Link cemented PFR (21)—Waldemar Link | Direct lateral approach |
| Shi et al. [25] | 31A1 (93)<br>31A2 (143) | PFNA—Synthes (236) | Instability (118)<br>Mechanical failure (87)<br>Both (31) | <2 years (154)<br>≥2 years (82) | UTA (116)<br>Corail uncemented stem (DePuy) and Continuum cup (Zimmer Biomet)<br><br>HTA (120)<br>Exeter stem (Stryker) and Continuum cup (Zimmer Biomet) | Not specified |

AVN: avascular necrosis; CMN: cephalo-medullary nail; MFS: modular femoral stem; UTA: uncemented total hip arthroplasty; CTA: cemented total hip arthroplasty; DM: dual mobility; PFR: proximal femoral replacement; HTA: hybrid total hip arthroplasty.

**Table 3.** MINORS criteria.

| Year | Author | MINORS | A Clearly Stated Aim | Inclusion of Consecutive Patients | Prospective Collection of Data | Endpoint Appropriate for Aim of Study | Unbiased Assessment of Study Endpoint | Follow-Up Period Appropriate | Loss of Follow-Up Less than 5% | Prospective Calculation of Study Size | Adequate Control Group | Contemporary Group | Baseline Equivalence of Groups | Adequate Statistical Analysis |
|---|---|---|---|---|---|---|---|---|---|---|---|---|---|---|
| 2020 | Godoy-Monzon [27] | 11 | 2 | 2 | 1 | 2 | 0 | 2 | 2 | | | | | |
| 2020 | Yu [26] | 21 | 2 | 2 | 1 | 2 | 0 | 2 | 2 | 2 | 2 | 2 | 2 | 2 |
| 2021 | Innocenti [24] | 10 | 2 | 2 | 1 | 2 | 0 | 2 | 1 | | | | 2 | |
| 2022 | Shi [25] | 20 | 2 | 2 | 1 | 2 | 0 | 2 | 1 | 2 | 2 | 2 | 2 | 2 |

### 3.3. Clinical Results

3.3.1. Functional Outcomes

Three studies [25–27] reported mean changes in the Harris Hip Score after conversion total hip arthroplasty; on the other hand, only one study [24] reported the Harris Hip Score after conversion partial and total hip arthroplasties without making a distinction regarding the procedure. When the preoperative Harris Hip Score was available, the salvage procedure determined an increase in this score, with mean value of 86.1 at an average of 2 years follow-up. Two studies [25,26] reported a longer follow-up for conversion total hip arthroplasties with a mean Harris Hip Score value of 88.5 after 5 years and a mean Harris Hip Score value of 82.1 after 10 years (Figure 2).

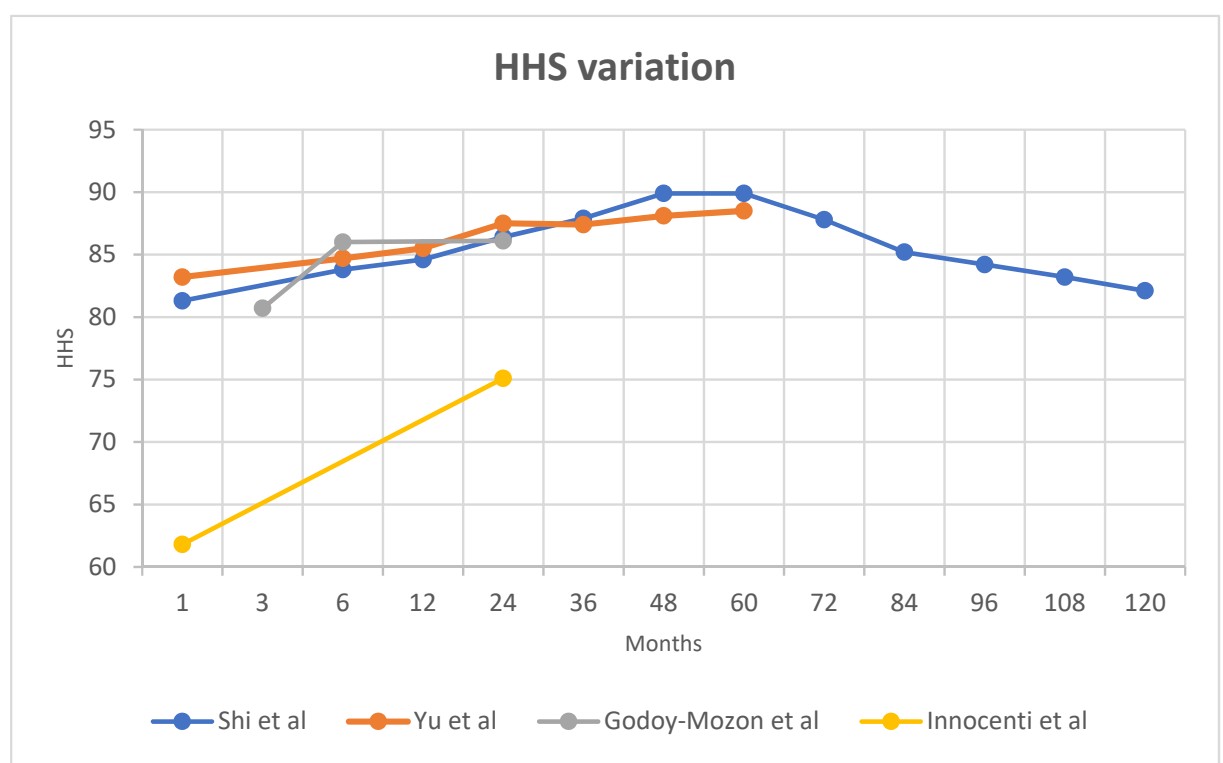

**Figure 2.** HHS variation during follow-up [24–27].

Shi et al. [25] reported the longest follow-up for conversion total hip arthroplasties after 10 years, with a mean Harris Hip Score of 78.6 for the uncemented implant group and a mean Harris Hip Score of 85.6 for the hybrid implant group (*p*-value 0.017) (Figure 3).

Yu et al. [26] reported the last follow-up after 5 years, with a mean Harris Hip Score of 87.6 for the uncemented implant group and a mean Harris Hip Score of 89.4 for the cemented implant group (*p*-value 0.021). This study also reported the difference in Harris Hip Score between the asymptomatic group and the symptomatic group at the time of surgery, considering also if the implants were cemented or uncemented. The mean Harris Hip Score value after 5 years was 90.2 ± 16.8 in the asymptomatic and cemented implant group, 88.6 ± 15.3 in the symptomatic and cemented implant group, 88.6 ± 15.2 in the asymptomatic and uncemented implant group, and 86.2 ± 14.2 in the symptomatic and uncemented implant group (*p*-value 0.031) (Figure 4).

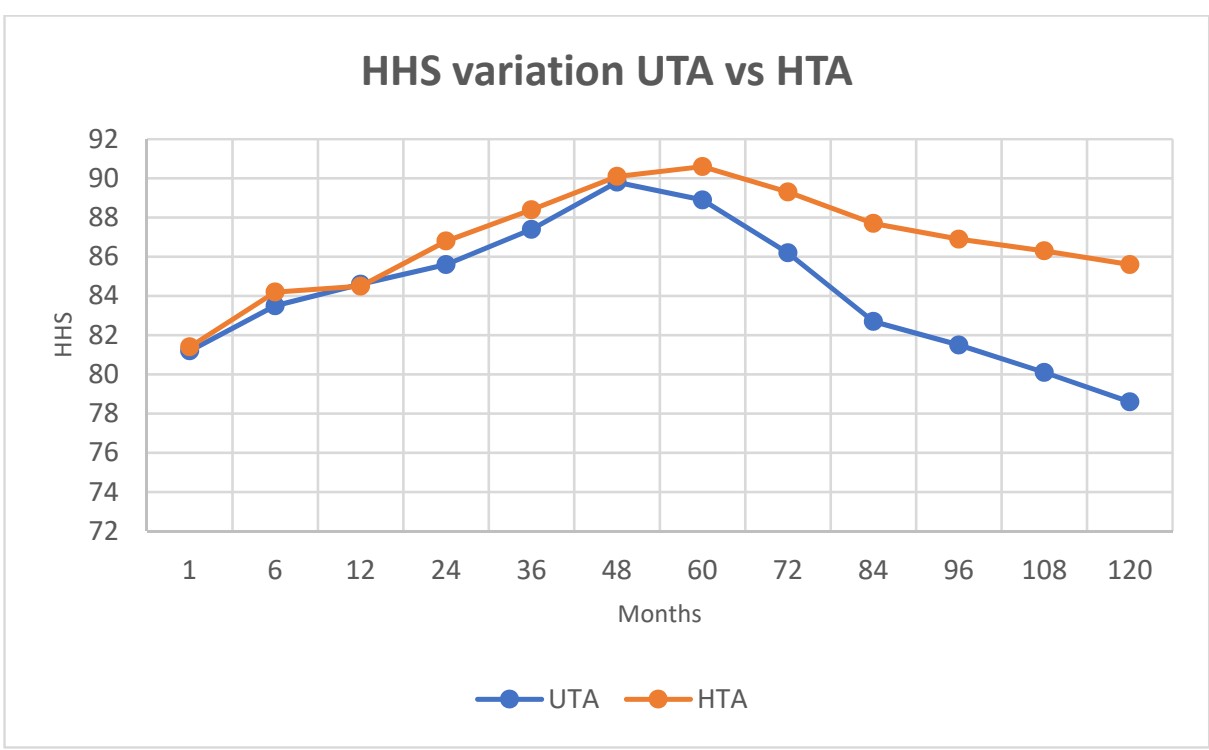

**Figure 3.** HHS variation differences among uncemented THA group and hybrid THA group.

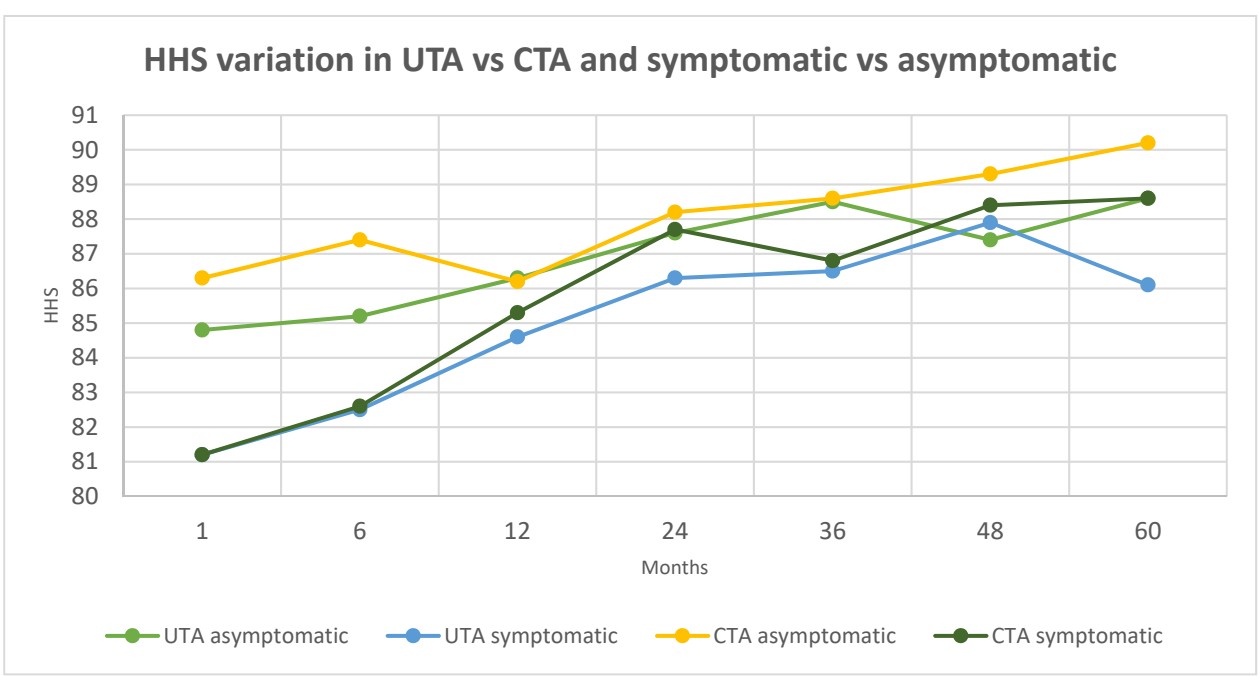

**Figure 4.** HHS variation differences between uncemented THA (UTA) group and cemented THA group (CTA).

3.3.2. Complications

All the selected studies reported the complications that occurred during follow-up, including dislocation, aseptic loosening, periprosthetic fracture and others (wound infection, abductor deficiency and intolerable hip pain) (Figure 5).

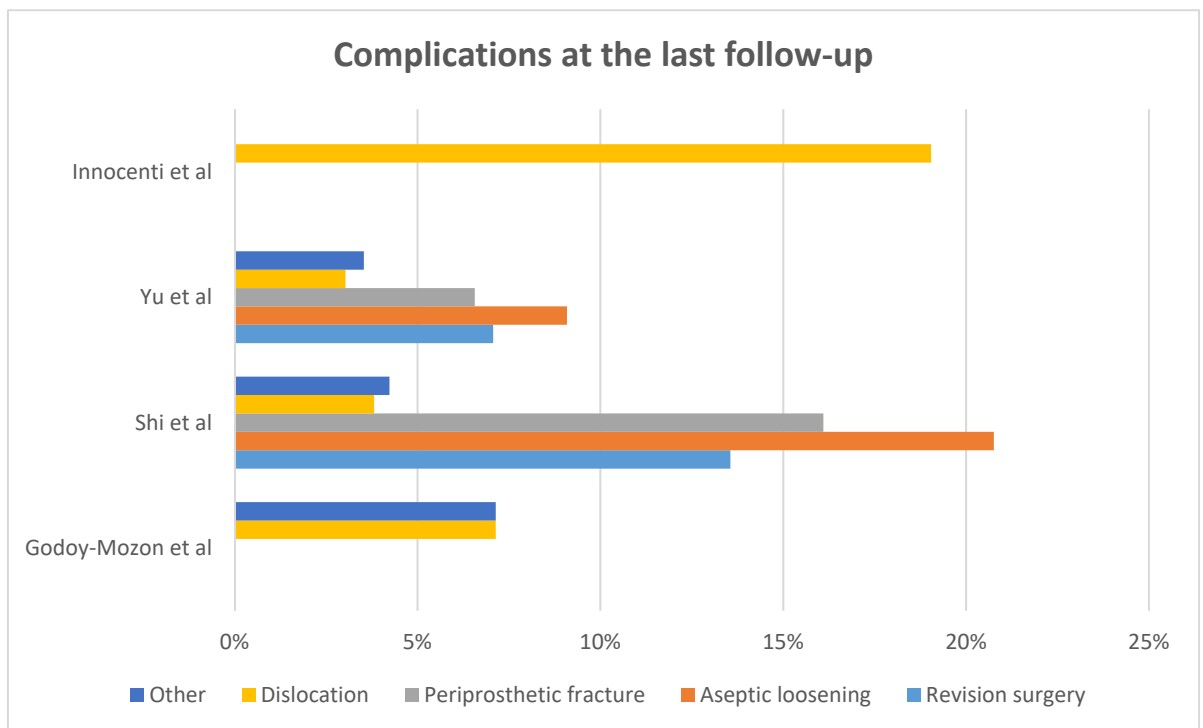

**Figure 5.** Complication rate at last follow-up [24–27].

Godoy-Monzon et al. [27] reported two dislocations (7%) treated with closed reduction and physical therapy, one superficial wound infection treated with antibiotic therapy and one abductor deficiency treated with physical therapy.

Shi et al. [25] classified complications depending on the type of implant. The uncemented total hip arthroplasty (UTA) group (116 patients) reported 94 complications in 39 patients: 23 revision surgeries (20%) (11 for aseptic femoral loosening and 6 for periprosthetic fracture), 32 cases of aseptic loosening (28%), 27 periprosthetic fractures (23%), 5 dislocations (4%) and 7 cases of intolerable hip pain (6%). The hybrid total hip arthroplasty (HTA) group (120 patients) reported 44 complications in 21 patients: 9 revision surgeries (8%) (of which 2 were for aseptic femoral loosening and 1 was for periprosthetic fracture), 17 cases of aseptic loosening (14%), 11 periprosthetic fractures (9%), 4 dislocations (3%) and 3 cases of intolerable hip pain (3%) (Figure 6). The revision rate in the uncemented group was higher than that in the hybrid group (*p*-value 0.006).

Yu et al. [26] also classified complications depending on the type of implant. The uncemented total hip arthroplasty group (98 patients) reported 40 complications: 23 revision surgeries (11%) (4 for aseptic acetabular loosening, 3 for aseptic femoral loosening, 1 for aseptic loosening of both components, 1 for dislocation, 1 for wear of the implant and 1 for periprosthetic fracture), 13 cases of aseptic loosening (13%), 10 periprosthetic fractures (10%), 3 dislocations (3%) and 3 cases of intolerable hip pain (3%). The cemented total hip arthroplasty (CTA) group (100 patients) reported 18 complications: 3 revision surgeries (3%) (2 for aseptic acetabular loosening and 1 for dislocation), 5 cases of aseptic loosening (5%), 3 periprosthetic fractures (3%), 3 dislocations (3%) and 4 cases of intolerable hip pain (4%) (Figure 7). The revision rate in the uncemented group was higher than that in the cemented group (*p*-value 0.025).

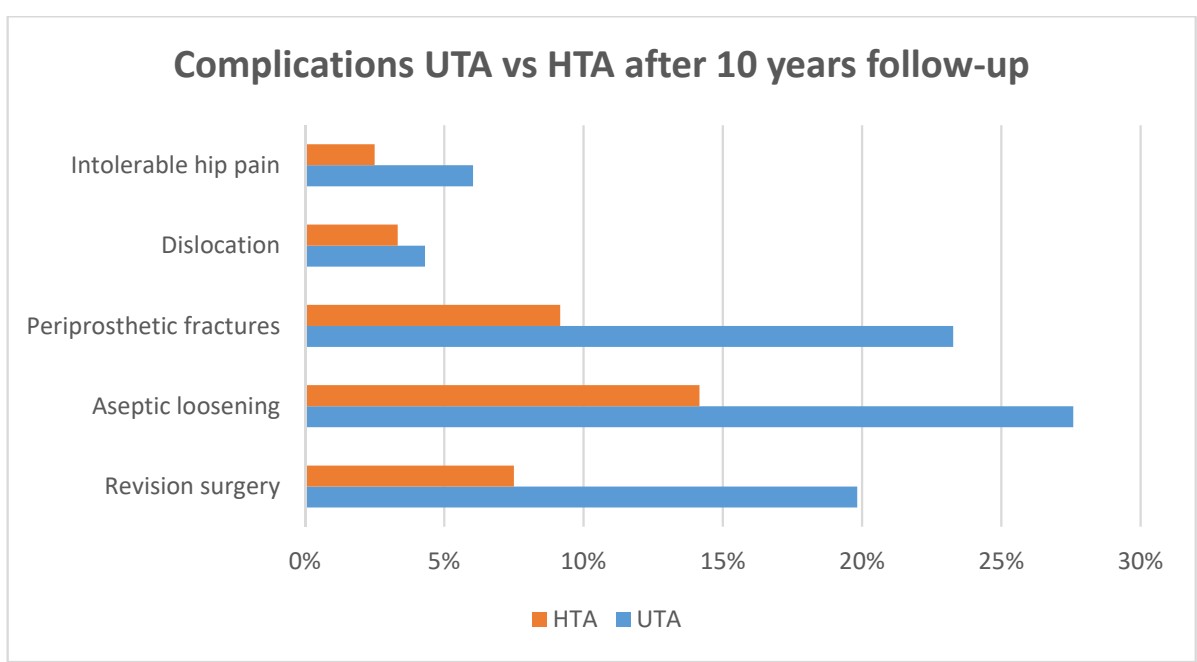

**Figure 6.** Complication rate difference between uncemented THA (UTA) group and hybrid THA group (HTA).

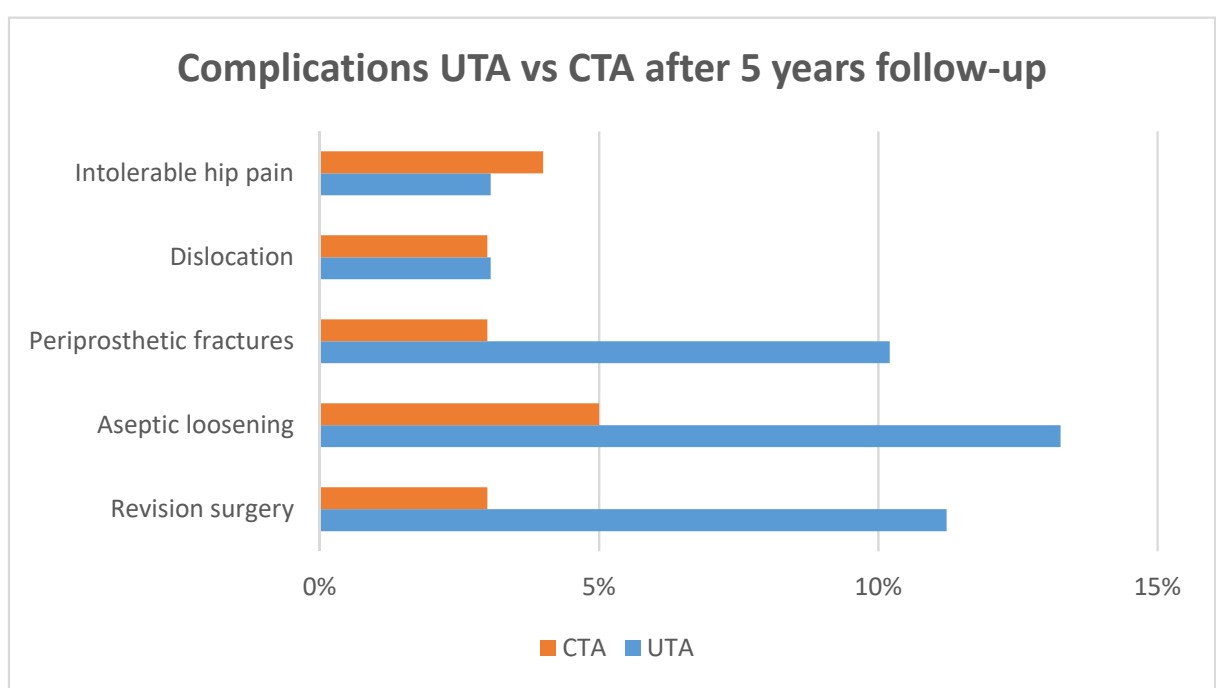

**Figure 7.** Complication rate difference between uncemented THA (UTA) group and cemented THA group (CTA).

Innocenti et al. [24] reported four complications, which were all dislocations (19%). Two patients underwent revision surgery: one had a conversion from hemiarthroplasty to dual-mobility total hip arthroplasty, and the other patient had a revision to change the 10° angled liner with a 20° angled liner. The other two dislocations were treated with closed reduction and physical therapy.

### 3.3.3. Survivorship

The survivorship of these implants was good on average (Table 2), but the causes of revision were various and depended on the specific type of implant.

Godoy-Monzon et al. [27] reported a revision-type implant survivorship of 100% after 5 years of follow-up, treating all the complications conservatively and successfully with good functional outcomes.

Yu et al. [26] compared the survivorship of two implant groups, 88.5% for the uncemented total hip arthroplasties and 97% for the cemented total hip arthroplasties after 5 years of follow-up ($p$-value 0.01).

Innocenti et al. [24] reported a proximal femoral replacement survivorship of 85% after 2 years of follow-up, where all revisions were due to a dislocation.

Shi et al. [25] compared the survivorship of two implant groups, 80.1% for the uncemented total hip arthroplasties and 92.5% for the hybrid total hip arthroplasties after 10 years of follow-up ($p$-value 0.004).

## 4. Discussion

Femoral nail failure is a quite rare event (4% to 17% in patients with pre-existing osteoporosis) [29], but it results in significant patient disability, and surgical treatment can be difficult and time-consuming due to the numerous technical issues, including unstable fractures, proximal femoral bone loss, acetabular erosion from lag screw cut-out, comminuted calcar and difficult implant removal, particularly when broken.

Fixation failure salvage surgery can be re-osteosynthesis or hip arthroplasty. The osteosynthesis revision is limited by a variety of parameters in the elderly, so conversion arthroplasty becomes the preferred treatment in these patients to give them better clinical and functional results [22].

This systematic review confirmed that conversion arthroplasty is an optimal treatment in femoral nail failure, allowing good hip function to be regained in the early postoperative period.

Hip function changes during the follow-up period, with an excellent result in hip function especially in the early follow-up period, and decaying in a longer follow-up period, probably connected to aging (Figure 2). The lack of data in the literature precluded the comparison between conversion total hip arthroplasties and conversion partial hip arthroplasties after nail failure: hemiarthroplasty is typically the preferred medical procedure for older individuals since it requires less time during surgery and results in less blood loss [30], but we do not have any information regarding patients that underwent conversion partial hip arthroplasties. Reviewing the outcomes of different conversion total hip arthroplasties, the hybrid implant group scored higher functional outcomes than the uncemented implant group, with an advantage trend year by year. Also, hip function was higher in the cemented implant group over the uncemented implant group, and at the same time, it showed an improved outcome in the group of patients who underwent conversion surgery in an asymptomatic status, compared with the group of patients who underwent conversion surgery after experiencing symptoms related to nail failure. This can be due to more effective physical therapy after conversion surgery and a better starting condition before surgery.

Several complications are reported in the literature, and there seems to be a higher complication rate in conversion total hip arthroplasty compared to primary total hip arthroplasty [31]. According to the literature, the most common is dislocation, which could be caused by previous surgical approaches, a weak hip abductor system or a lack of medial offset [32]. Despite the literature, the second finding of this systematic review is that, overall, the main complication in conversion total hip arthroplasty consists of aseptic loosening of the implant. Aseptic loosening, such as periprosthetic fracture, is a severe complication that depends more on the type of implant and the implant–bone interface than other variabilities. The higher rate of complications in uncemented total hip arthroplasties may be due to microfracture and bone remodeling during the postoperative period that could lead to

a major rate of aseptic loosening and periprosthetic fracture and eventually to revision surgery (Figures 6 and 7). However, the collected complication data are underestimated because studies including a wider population considered revision surgery as a complication, including in this definition the causes that led to revision arthroplasty surgery. Other complications that were not included have been treated with a non-surgical approach, probably in relation to the elderly and more fragile population involved.

Two authors [24,27] have chosen revision implants (modular femoral stems and proximal femoral replacement) instead of primary implant design, thus achieving hip balance and stability without the incidence of component failure. These studies have showed the same outcome in terms of Harris Hip Score and hip anatomic restoration during the years but differed from the other studies by the type of complications: no aseptic loosening or periprosthetic fractures were recorded.

Two studies [25,26] in our review, in particular, have showed the same outcome in terms of survivorship of the implants: at a 3-year follow-up, survivorship between the two groups in the analysis (uncemented vs. cemented total hip arthroplasties and uncemented vs. hybrid total hip arthroplasties) seem to be comparable, but it starts to differ over time; in fact, uncemented total hip arthroplasties provides inferior long-term survivorship than hybrid and cemented total hip arthroplasties. The good overall survivorship of these implants shows that this is a worthy procedure to offer to the elderly patient; however it is important to choose the right implant and fixation considering the patient and the femoral bone quality.

There are some limitations in the current study. The studies included in our analysis were only four, and all had a retrospective design. Many studies had to be excluded because they did not satisfy the eligibility criteria; in particular, several studies reported other types of scores instead of the most common Harris Hip Score, included patients treated with different means of femoral fracture fixation or had insufficient details. Multiple surgeons performed surgeries in various groups of patients, using different surgical approaches and implants. Even if all studies had a minimum follow-up period of 2 years, a longer follow-up period needs to be continued to understand the long-term survival of these salvage procedures.

In addition, it may be useful to introduce the SF-36 Health Status Scale into the evaluation system in future studies to evaluate quality of life for patients before and after conversion total hip arthroplasty as well as hip function with the Harris Hip Score.

## 5. Conclusions

In conclusion, conversion arthroplasty is a successful salvage treatment for femoral nail fixation failure in the elderly population. The present study found no big differences in short-term functional outcome between uncemented, cemented and hybrid total hip arthroplasties, even if the hybrid or cemented total hip arthroplasty groups had a better outcome than the uncemented total hip arthroplasty groups in the long term, as the revision rate suggests. Further investigations are still needed to provide a definitive answer on the best way to treat a femoral nail fixation failure in terms of surgical timing and approach, the type of implant and patient outcomes.

**Funding:** This research received no external funding.

**Institutional Review Board Statement:** An IRB Statement was not necessary for the nature of this study.

**Informed Consent Statement:** Patient consent was not necessary because the data used in this study were already published.

**Data Availability Statement:** No new data were created or analyzed in this study. Data sharing is not applicable to this article.

**Conflicts of Interest:** All authors declare that they have no known competing financial interests or personal relationships that could have appeared to influence the work reported in this paper.

Therefore, no benefits in any form have been received or will be received from a commercial party related directly or indirectly to the subject of this article.

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
