# Peer review of "Salvage Hip Arthroplasty in Nail Failure: A Systematic Review"

_prosthesis, doi:10.3390/prosthesis5040092_

Round 1

Reviewer 1 Report

Comments and Suggestions for Authors

The authors of this article have performed a systematic review “to evaluate functional outcomes and complications of arthroplasty after CMN failure”. Although the paper is of interest, I have some considerations:

1.       The authors have not declared the aim of the study in the abstract.

2.       The authors used THA, PHA, and HHS abbreviations in the abstract without defining them first.

3.       The authors have divided the introduction into so many paragraphs, which is quite unusual and unacademic. Each paragraph should have an intro, body, and a conclusion at the end. Please edit accordingly.

4.       There is no clear literature gap highlighted in the introduction.

5.       The aim of the study is very broad. For example, which functional outcomes are going to be evaluated? Which population? Is there a comparator? Is there a timeframe? Is there a specific setting?

6.       There are many missing details in the methods section:

a.       The search terms.

b.       Manual search details.

c.       Quality assessment.

d.       Qualitative/quantitative synthesis methods.

7.       What do the authors mean by including only level III or IV evidence. Please explain more what is that as it may differ between guidelines and also tell us why is that?

8.       Again, what clinical outcomes are we looking at here? The research question, aim, inclusion/exclusion criteria should explicitly indicate the targeted clinical outcomes.

So, the main limitations in this study are related to the unclear research methodology implemented in the conduction of this study. Not only that, but there are some areas that the manuscript has made different from the protocol submitted to PROSPERO. For example, the manuscript says the authors included articles of level III or IV evidence, but the protocol clearly states I to III. Another point is serious in the risk of bias (quality) assessment, where the authors said that “the studies included will be a level of evidence I to III. Two reviewers will be involved and the third reviewer will decide on every disagreements”. What does that mean? This means that the guidelines and the proper conduction of systematic reviews have nt been followed.

Comments on the Quality of English Language

Average

Author Response

Thank you for the review and for your important and precise notes.

Please see the attachment below.

Reviewer 2 Report

Comments and Suggestions for Authors

well done

Author Response

Thank you for the review

Reviewer 3 Report

Comments and Suggestions for Authors

Thanks to the authors for the interesting topic argued. Proximal lateral hip fractures are very current in elderly population, and a good assessment and treatment is the main aim of surgeons.

Manuscript structure is properly organized: it results complete in all its section, and clear in its content. The abstract is comprehensive ad offers a clear overview of the study. The methodological approach is correct. Charts and data are well reported.

Authors could specify the acronym HHS.

Results are in line with the aim of the study and provide important practical advice.

The study presents also many limitations, such the short-medium follow-up and the limited number of the studies. I would like to encourage the authors to pursue additional research in order to overcome these limitations.

Author Response

Thank you for the review and your important notes.

HHS stands for Harris Hip Score and I'll update it in the revision of the manuscript.

Unfortunately the main limit of the study is the poor number of patients and the short follow-up. There are very few studies in literature about the specific treatment of nail failure with hip arthroplasty; there is a larger amount of papers on the failure of "fracture fixation" in the proximal femur but they involve other means of fixation as plates and screws, so it makes difficult to analyze the specific outcome of nail failure with more details. For this reason we had to reduce the number of the included studies in our review. We hope to see more studies on this topic in the next years to improve these results.

Reviewer 4 Report

Comments and Suggestions for Authors

This review article is clinically important as it systematically investigates the clinical results of conversion arthroplasty for the femoral nail fixation failure. However, the description is not kind to the readers, which should be modified, readable and easy to understand. Please consider the following review comments.

The author uses many abbreviations, which makes the readers confused. At the first appearance of the abbreviation, the definition should be clearly described in the main text. Some of those are described only in the table captions underneath of Tab.1 and 2. Some others are noted only abbreviation without properly explanation i.e. HHS and cTHA. What is the difference between UTA and cTHA? It seems that both are the same as the cemented Total Hip Arthroplasty.

Regarding Fig.1 and 2, the unit is required for the horizontal axis. It seems the unit is months, but the main text described the followup period in years. Also the vertical axis needs the unit of point or score. 

The subheading of 3.3. Subsubsection is not appropriate, 3.3. Clinical evaluations or Clinical results is recommended. Also the contents list of 3 lines is not necessary.

Regarding 4 articles subjected to this study, are they from the same country? Since the online search has performed worldwide, the articles seem to be collected from various countries. As additional information, the country should be described in order to consider the difference of lifestyle and medical environment.

In Discussion, the phrase ‘The first findings’ is an exaggerated expression. The author already concluded it in the above paragraph. The paragraph should be simply changed to ‘this systematic review confirmed that conversion arthroplasty is an optimal treatment in femoral nail failure allowing …….’ After this paragraph, the discussions are exactly important to derive the conclusion. The significance of this review article is as the author described in Conclusions.

Author Response

Thank you for the review and for your important notes.

I'll use them to improve the manuscript during revision, especially the use of abbreviations and the units in the figures.

Regarding the 4 studies selected, 2 have been conducted in China, 1 in Italy and 1 in Argentina, but we don't believe that this has influenced the choice of treatment and the approach to the patient.

Reviewer 5 Report

Comments and Suggestions for Authors

This manuscript reviewed salvage arthroplasty after ORIF failure for hip fracture. Total four papers were  adopted  and revealed that cementless THA had poorer results than hybrid one.

I have a few minor comments, explained below.

P8, last line: 3.1. Subsection----delete

P9, Table 1 CMN: cephalon----cephalo-

References: Please abbreviate the journal names

#5,6,8,10,11,12,14,15,23,28,29,31,32.

Author Response

Thank you for the review and your notes.

I'll use them to improve the manuscript during the revision.

Round 2

Reviewer 1 Report

Comments and Suggestions for Authors

The authors have improved the manuscript significantly. No further comments. 

Comments on the Quality of English Language

No comments.